# Engineered clearing agents for the selective depletion of antigen-specific antibodies

Siva Charan Devanaboyina[1,2], Priyanka Khare[1,2], Dilip K. Challa[1,2], Raimund J. Ober[1,3] & E. Sally Ward[1,2]

Here we have designed a novel class of engineered antibody-based reagents ('Seldegs') that induce the selective degradation of antigen-specific antibodies. We demonstrate the rapid and specific clearance of antibodies recognizing the autoantigen, myelin oligodendrocyte glycoprotein and tumour target, HER2. Seldegs have considerable potential in multiple areas, including the treatment of antibody-mediated autoimmunity and diagnostic imaging.

[1] Department of Molecular and Cellular Medicine, Texas A&M University Health Science Center, 469 Joe H. Reynolds Medical Sciences Building, 1114 TAMU, College Station, Texas 77843, USA. [2] Department of Microbial Pathogenesis and Immunology, Texas A&M University Health Science Center, 3107 Medical Research & Education Building, 8447 State Highway 47, Bryan, Texas 77807, USA. [3] Department of Biomedical Engineering, Texas A&M University, 5045 Emerging Technologies Building, 3120 TAMU, College Station, Texas 77843, USA. Correspondence and requests for materials should be addressed to R.J.O. (email: raimund.ober@tamu.edu) or to E.S.W. (email: sally.ward@tamu.edu).

The development of strategies to specifically decrease antigen-specific antibody levels for the clearance of deleterious antibodies during therapy and diagnosis represents a longstanding and unresolved challenge. Such an approach would have broad utility in areas such as the treatment of antibody-mediated autoimmunity, antibody-mediated transplant rejection and the clearance of background during diagnostic imaging. The knowledge that the neonatal Fc receptor, FcRn, maintains immunoglobulin G (IgG) levels and transport in the body has prompted the development of antibody- or peptide-based inhibitors to reduce IgG levels[1–5]. However, these inhibitors block the interaction of the Fc region of IgG with FcRn and decrease levels of IgGs of all specificities, including protective antibodies. To overcome these off-target effects, here we have designed a novel class of engineered antibodies that selectively clear antigen-specific antibodies without modulating the levels of antibodies of other specificities. We have named these agents 'Seldegs' to indicate their ability to selectively degrade antibodies of defined specificities.

The design of clearing agents for antigen-specific antibodies presents several challenges: first, antigen-specific antibody levels are typically very low compared with those of antibodies of irrelevant specificities, and these two antibody pools are similar with shared constant regions but distinct variable domains. Second, since the antibodies being targeted are bivalent, cross-linking could result in inflammatory immune complexes. Seldegs were therefore designed to display recombinant antigen as a monomer linked to a dimeric, human IgG1-derived Fc fragment using a similar approach to that described previously for monomeric erythropoietin (Epo)-Fc fusions[6]. Mutations to ablate interactions with human FcγRs[7] and enhance binding to FcRn in the pH range 6.0–7.4 (ref. 1) were inserted. Naturally occurring IgGs have substantially higher affinity for FcRn at acidic pH than at near neutral pH, and this property is essential for the recycling and transcytosis of IgG within FcRn-expressing cells[8]. By contrast, gain of binding affinity of an Fc (or IgG) for FcRn at pH 7.4 results in receptor-mediated internalization into cells and lysosomal delivery[1,9,10]. In the current study, we demonstrate that Fc-antigen fusions, or Seldegs, containing such affinity-enhanced Fc fragments selectively capture antigen-specific antibodies and direct them into degradative lysosomal compartments in FcRn-expressing cells.

## Results

**Clearance of antigen-specific antibodies by Seldegs.** To demonstrate the generality of the Seldeg approach, antibodies specific for two antigens were targeted (Fig. 1a): first, the extracellular domain (ECD) of myelin oligodendrocyte glycoprotein (MOG-Seldeg), which is recognized by autoreactive antibodies in both animal models of multiple sclerosis and multiple sclerosis in humans[11–13]. Second, the ECD of HER2 (HER2-Seldeg), a well-defined target for therapy and diagnostic imaging of HER2-overexpressing tumours with HER2-specific antibodies such as trastuzumab (TZB)[14]. Both MOG- and HER2-Seldegs have the expected binding properties for FcRn, MOG-specific antibody (8-18C5 (ref. 15)) and TZB (Fig. 1b; Supplementary Fig. 1; Supplementary Table 1). The expression yields of the Seldegs are ∼50 mg l$^{-1}$ (MOG-Seldeg) and ∼15 mg l$^{-1}$ (HER2-Seldeg). SDS–polyacrylamide gel electrophoresis and high-performance liquid chromatography analyses also indicate that these Fc fusion proteins have favourable biophysical properties following storage at 4 °C (30 days) or 37 °C (5 days) in PBS or human serum (Supplementary Fig. 2). In addition, the Seldegs retain their affinity for binding to 8-18C5 ($K_D = 33.0$ nM; MOG-Seldeg) and TZB ($K_D = 14.3$ nM; HER2-Seldeg) following incubation for 5 days at 37 °C.

We first investigated the ability of Seldegs to clear antigen-specific antibodies in transgenic mice that express human FcγRs (huFcγR mice)[16]. These mice were used since Seldegs have human IgG1-derived Fc regions. Mice were injected with $^{125}$I-labelled, MOG-specific antibody 8-18C5 (ref. 15). Twenty-four hours later, MOG-Seldeg was injected at a 16-fold (high dose) or 4-fold (low dose) molar excess over target. The delivery of MOG-Seldeg resulted in a rapid decrease in 8-18C5 levels in the blood and whole body (Fig. 1c). Importantly, the total IgG levels in serum of mice before and 48 h following treatment with high or low dose of Seldeg were not significantly different (Supplementary Fig. 3), indicating the selectivity of Seldeg-mediated clearance. In addition, injection of an analogous construct without the FcRn-enhancing MST-HN mutations (MOG-WT) had no effect on 8-18C5 clearance (Fig. 1c).

To further analyse the specificity of Seldegs and their effect on antibodies with different antigen recognition properties, the behaviour of the HER2-specific humanized antibody TZB[14] was investigated in the presence of HER2-Seldeg and MOG-Seldeg (Fig. 1d). HuFcγR mice were injected with $^{125}$I-labelled TZB and subsequently with a 4-fold molar excess of Seldeg. As controls, equivalent molar amounts of HER2-WT (analogous to HER2-Seldeg without FcRn-enhancing mutations), MOG-Seldeg or hen egg lysozyme-specific human IgG1 with the MST-HN mutations (Abdeg)[1] were used. This Abdeg contains the same mutations to increase FcRn binding as those in the Seldegs. HER2-Seldeg induced a decrease in TZB levels in the blood and whole body, whereas the control proteins resulted in similar behaviour to that observed for vehicle (PBS; Fig. 1d). Remarkably, blood levels of TZB were reduced to ∼30% of the injected dose within 2 h following Seldeg delivery (Fig. 1e).

Treatment with MOG- and HER2-Seldeg resulted in biphasic clearance of the targeted radiolabelled antibodies (Fig. 1c,d). Within the first ∼50 h of Seldeg delivery, the radiolabelled antibodies are rapidly cleared. This is followed by slower clearance rates that are close to those in control animals (Fig. 1d). Analyses of the pharmacokinetic behaviour of MOG- and HER2-Seldeg revealed relatively short β-phase half-lives (MOG-Seldeg, 47.0 ± 1.4 h (s.d.); HER2-Seldeg, 38.3 ± 3.1 h; $n = 5$ for both groups) that were similar to those of the Abdeg (37.8 ± 2.5 h; $n = 5$; Supplementary Fig. 4). For comparative purposes, we also determined the β-phase half-lives of MOG- and HER2-WT (102.3 ± 22.0 and 87.2 ± 11.9 h, respectively; $n = 5$ for both groups). The rapid clearance of Seldegs is consistent with earlier studies demonstrating that engineered antibodies with increased binding to FcRn at near neutral pH results in shorter *in vivo* half-lives[8,17,18]. This behaviour indicates that the persistence of a low percentage of 8-18C5 or TZB following ∼50 h of treatment is due to reduced levels of Seldeg, resulting in incomplete capture of the targeted antibodies.

**Seldegs internalize targeted antibodies into lysosomes.** To investigate the mechanism of Seldeg activity at the cellular level, we analysed their effects on the internalization and accumulation of 8-18C5 and TZB in endothelial cells (HMEC-1, human microvasculature endothelial cells) transfected with a human FcRn-GFP expression construct (mutated to confer binding properties analogous to those of mouse FcRn[19]). Seldegs efficiently internalize cognate antibodies into endosomes within cells (Fig. 2a–c). Importantly, high levels of 8-18C5 or TZB in cells result from treatment with MOG- or HER2-Seldeg, respectively, and the majority of targeted antibody is retained by the cells during a 60 min chase (Fig. 2a). By contrast, target antibody accumulation within cells is substantially lower in the presence of MOG-WT (8-18C5) or HER2-WT (TZB), and the

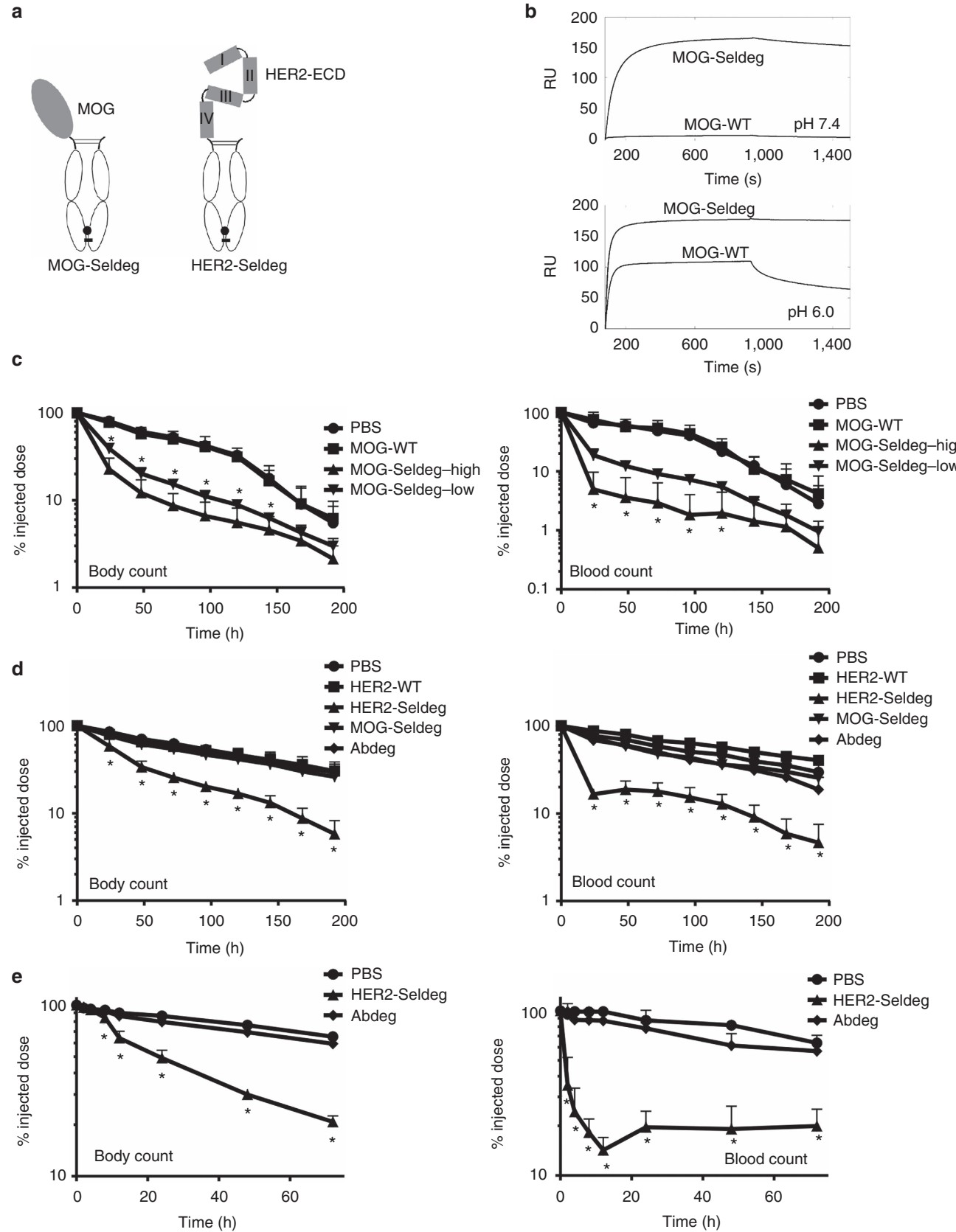

internalized antibody is efficiently recycled over 60 min (Fig. 2a). Following 8 h incubation, the targeted antibody and corresponding Seldeg are delivered to lysosomes (Fig. 2d; Supplementary Fig. 5). Importantly, cells do not accumulate 8-18C5 and TZB in lysosomes in the presence of HER2-Seldeg or MOG-Seldeg, respectively (Fig. 2d; Supplementary Fig. 5).

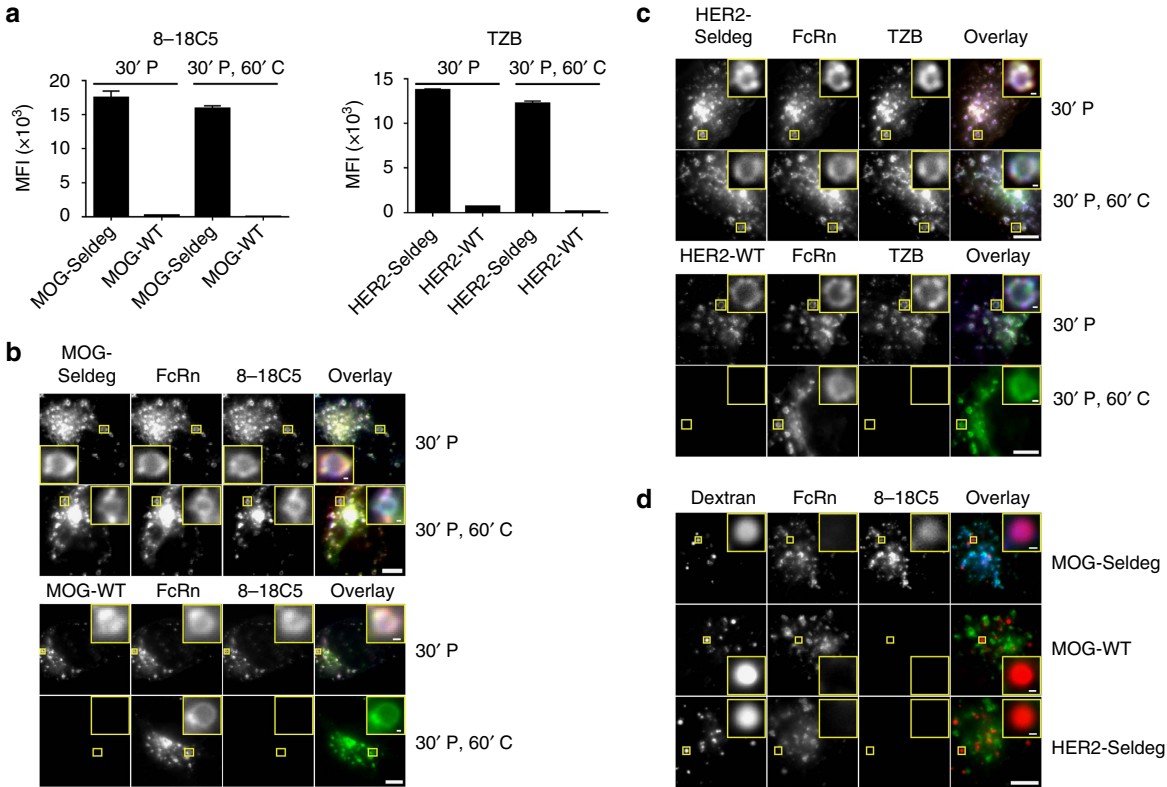

**Figure 2 | Seldegs increase the accumulation of antigen-specific antibodies in human endothelial (HMEC-1) cells expressing FcRn-GFP. (a)** HMEC-1 cells were incubated with 100 nM Alexa 647-labelled 8-18C5 (MOG-specific) or TZB (HER2-specific) in complex with 400 nM MOG-Seldeg/MOG-WT or HER2-Seldeg/HER2-WT for 30 min and chased for 0 (30′ P) or 60 min (30′ P, 60′ C). Mean fluorescence intensities (MFI) of Alexa 647-labelled 8-18C5 or TZB for triplicate samples were determined by flow cytometry. Error bars indicate s.d. **(b,c)** HMEC-1 cells were plated on coverslips and treated as in **a**, except that Seldegs or control WT proteins were labelled with Alexa 555 and cells were fixed for microscopy. Images of representative cells from multiple cells analysed are shown with GFP, Alexa 555 and Alexa 647 in overlays pseudocoloured green, red and blue, respectively. Representative endosomes in the insets are cropped and expanded. **(d)** HMEC-1 cells were pre-pulsed with Alexa 555-labelled dextran for 2 h, washed and pulsed with 8-18C5 in complex with MOG-Seldeg, MOG-WT and HER2-Seldeg (concentrations and labels as for **a**) for 30 min, followed by an 8 h chase. Cells were washed, fixed and imaged, and images for a representative cell from multiple cells analysed are presented. Representative lysosomes in the insets are cropped and expanded. For the overlay, GFP, Alexa 555 and Alexa 647 are pseudocoloured as in **b**. For **b-d**, scale bars = 5 μm, and for insets, scale bars = 0.25 μm. Data shown are representative of at least two independent experiments.

In summary, we have demonstrated that engineered antigen-Fc fusion proteins (Seldegs) induce rapid and substantial decreases of antigen-specific antibody levels. Further, Seldegs are effective at relatively low doses that, by contrast with earlier approaches[1–5], do not alter the clearance of antibodies of non-targeted specificities. The properties of Seldegs indicate their potential applications in multiple clinical situations where it is desirable to selectively clear antibodies of defined specificities.

## Methods

**Cell lines.** HMEC-1 cells, a generous gift from Francisco Candal (CDC), were maintained in phenol red-free HAMS F12-K medium (US Biological). BT-474 (American Type Culture Collection; HTB-20) were maintained according to the

supplier's protocol. Cell lines were tested at monthly intervals for mycoplasma contamination and were authenticated by short tandem repeat analysis.

**Generation of expression constructs.** The gene encoding the HER2 leader peptide and ECD (consisting of 629 residues) was isolated from a HER2-over-expressing breast cancer cell line (BT-474) employing standard molecular biology techniques. This gene was fused via a IEGRMD linker peptide to the N terminus of the hinge region of a gene encoding the human IgG1-derived Fc fragment using splicing by overlap extension[20]. Mutations to ablate binding to FcγRs (G236R/L328R; EU numbering)[7], enhance binding to FcRn (MST-HN; M252Y/S254T/T256E/H433K/N434F)[1] and generate 'knobs-into-holes' (Y349T/T394F)[21,22] were inserted into the Fc fragment gene using standard methods. Cysteine (C220) in the hinge region that bridges with cysteine in the light-chain constant region was also mutated to serine. Fc fragment genes without fused antigen were generated with complementary knobs-into-holes mutations (S364H/F405A)[21,22]. Similar methodology as described above was used to produce a fusion construct encoding

**Figure 1 | TZB and 8-18C5 are rapidly cleared by Seldegs in transgenic mice expressing huFcγRs. (a)** Schematic representation of Seldeg design. **(b)** A concentration of 100 nM MOG-Seldeg or MOG-WT was injected over immobilized mouse FcRn at the indicated pH. **(c)** Mice were intravenously injected with radiolabelled ([125]I) 8-18C5 (15 μg) and 24 h later PBS, MOG-WT (31 μg) or MOG-Seldeg (4-fold (31 μg) or 16-fold (125 μg) molar excess; low or high dose, respectively) was delivered intravenously. Radioactivity levels were determined at the indicated times. Whole body or blood levels obtained immediately before Seldeg or control delivery were taken as 100%. **(d,e)** The same methodology as in **c** was used, except that radiolabelled TZB (15 μg) injection was followed 24 h later by intravenous delivery of PBS, HER2-WT (51 μg), HER2-Seldeg (51 μg), MOG-Seldeg (31 μg) or Abdeg (MST-HN; 60 μg), each at fourfold molar excess. Error bars indicate s.d. and statistically significant differences are indicated for MOG-WT versus MOG-Seldeg (low) **(c)** HER2-Seldeg versus MOG-Seldeg **(d)** and HER2-Seldeg versus Abdeg **(e)** by *(P < 0.05, two-way analysis of variance with Tukey's multiple comparisons; n = 6 mice per group). Data shown are representative of two independent experiments.

the ECD of mouse MOG[23] linked to the same engineered Fc fragment, using a GGGGS linker. Analogous ('wild type') expression constructs were also made to express antigen-Fc fusions without the MST-HN mutations. Sequences of the expression constructs are available upon request.

**Protein expression and purification.** Recombinant proteins were expressed in HEK-293F (Life Technologies) cells following transient transfection with the Gibco Expi293 expression system kit (Life Technologies). The MST-HN mutations reduce binding of the Fc region to protein G-Sepharose and Seldegs were therefore purified using an anion exchange column (SOURCE-15Q, GE Healthcare) at pH 8.0 and a linear salt gradient (0–0.5 M NaCl). HER2-WT and MOG-WT were purified using protein G-Sepharose (GE Healthcare). 8-18C5 was expressed in recombinant form and purified using protein G-Sepharose[23], and clinical grade TZB (Herceptin; Roche) was obtained from the UT Southwestern Medical Center Pharmacy. Recombinant Abdeg (MST-HN, hen egg lysozyme-specific) was purified from culture supernatants using lysozyme-Sepharose[1]. All recombinant proteins were purified using size-exclusion chromatography (GE Healthcare) in PBS (Lonza) before use in experiments.

**Analyses of Seldeg stability.** For serum stability assays, endogenous IgGs were depleted from human male AB plasma (Sigma) by passage through protein G-Sepharose (GE Healthcare). Seldegs were incubated in serum at a concentration of 100 nM at 37 °C for 3 or 5 days. Following incubation, Seldegs were immuno-precipitated using agarose beads coupled to goat anti-human Fc-specific antibody (Sigma). Immunoprecipitated Seldegs were run on 12% SDS–polyacrylamide gels, transferred to polyvinylidene difluoride membranes (Millipore) and membranes incubated with horseradish peroxidase-conjugated goat anti-human Fc-specific (H + L) antibody (Jackson ImmunoResearch). Bound secondary conjugate was detected using Westernsure substrate, followed by scanning with a C-DiGit blot scanner (LI-COR).

Seldegs were also incubated in PBS (Lonza) at 4 °C (30 days) or 37 °C for 5 days, followed by analyses using a Superdex 200 5/150 GL column (GE Healthcare), 12% SDS–polyacrylamide gel electrophoresis and surface plasmon resonance (BIAcore).

**Surface plasmon resonance analyses.** Surface plasmon resonance experiments were carried out using a BIAcore T200 (GE Healthcare). To determine the equilibrium binding affinities for the interactions of Seldeg/WT proteins with antibodies (8-18C5 or TZB), 8-18C5 or TZB were injected over immobilized Seldeg/WT proteins (coupled at ~350–2,000 relative units on flow cells of CM5 sensor chips) at a flow rate of 10 µl min$^{-1}$ in PBS (Lonza; pH 7.4) plus 0.01% v/v Tween-20. Flow cells were regenerated at the end of each run using 0.15 M NaCl and 0.1 M glycine (pH 2.3). The equilibrium dissociation constants ($K_D$s) for the interactions of Seldegs/WT with TZB or 8-18C5 were determined using a 1:1 interaction model and custom-written software[24]. Binding of Seldeg/WT to recombinant human and mouse FcRn in PBS (pH 6.0 or 7.4) plus 0.01% v/v Tween-20 was analysed by injecting 100 nM Seldeg/WT over immobilized FcRn (coupled at ~600 relative units on a CM5 sensor chip) at a flow rate of 10 µl min$^{-1}$ (ref. 19). The flow cells were regenerated using 0.15 M NaCl and 0.1 M sodium bicarbonate (pH 8.5).

**Antibody labelling.** 8-18C5, TZB, Seldeg, corresponding WT fusion proteins and MST-HN Abdeg were radiolabelled with $^{125}$I using Iodogen (Perkin Elmer or MP Biomedicals)[25]. 8-18C5 and TZB were fluorescently labelled with Alexa 647 Fluor (ThermoFisher Scientific) with antibody:dye ratios of 1.6 and 3, respectively. Seldeg/WT fusion proteins were fluorescently labelled with Alexa 555 Fluor (ThermoFisher Scientific) dye at a protein:dye ratio of 2.7 (MOG-WT), 2.5 (MOG-Seldeg), 1.5 (HER2-WT) and 1.2 (HER2-Seldeg) using the manufacturer's protocol.

**Fluorescence microscopy and recycling assay.** HMEC-1 cells were transiently co-transfected using Nucleofector technology (Lonza) with human FcRn-GFP (FcRn tagged at the C terminus with GFP) and human $\beta_2$ microglobulin[26]. FcRn-GFP containing, a mutated version of human FcRn with similar binding properties as mouse FcRn[19], was used throughout these studies.

For microscopy studies, FcRn-GFP-transfected cells were plated in IgG-depleted phenol red-free HAMS F-12K medium on micro-coverglasses (Electron Microscopy Sciences). Eighteen hours post transfection, cells were pulsed with labelled antibodies at a 1:4 molar ratio of 8-18C5 or TZB (15 µg ml$^{-1}$) to MOG- or HER2-Seldeg/WT (31 and 51 µg ml$^{-1}$ for MOG- and HER2-Seldeg/WT, respectively) for 30 min. This pulse was followed by two washes with ice-cold PBS and fixation with 1.7% (w/v) paraformaldehyde (Electron Microscopy Sciences) with 0.025% (v/v) glutaraldehyde (Sigma) on ice for 10 min. Alternatively, following the 30 min pulse, cells were washed with PBS and chased in medium for 60 min before further washes and fixation. To label lysosomes, transfected cells were pre-pulsed with Alexa 555-labelled dextran (ThermoFisher Scientific; 500 µg ml$^{-1}$) for 2 h, followed by pulsing with labelled antibodies (as above) for 30 min and a chase period of 8 h. All pulse-chase experiments were carried out in phenol red-free HAMS F-12K medium (pH 7.4) at 37 °C in a 5% $CO_2$ incubator.

Fixed cells were imaged with a Zeiss Axiovert 200M inverted microscope, and acquired data were analysed using in-house written software (MIATool)[26]. For recycling assays, transfected HMEC-1 cells were plated in 24-well plates at a density of 50,000 cells per well. Eighteen hours post transfection, cells were pulsed with medium containing labelled antibodies as for the microscopy experiments. At the end of the chase period, cells were washed with ice-cold PBS, followed by trypsinization (PBS and trypsin were maintained at pH 6.0). Trypsinized cells were collected, fixed on ice for 5 min with 1.7% paraformaldehyde (w/v) and analysed using a LSRFortessa flow cytometer (BD Biosciences). Flow cytometry data were analysed using Flowjo (FlowJo).

**Studies in mice.** All pharmacokinetic experiments were carried out in 8–10-week-old male or female C57BL/6 mice that transgenically express human FcγRs[16]. Animals were housed in a pathogen-free facility at Texas A&M University and all animal experiments were approved by the Texas A&M Institutional Animal Care and Use Committee. Mice were injected intravenously with 15 µg radiolabelled antibody (8-18C5 or TZB) in 200 µl 0.1% BSA in PBS (Lonza). Twenty-four hours later, Seldegs and controls ($n = 6$ mice per group) were intravenously delivered in 200 µl PBS at a 4-fold or 16-fold molar excess as indicated in the figure legends. Whole-body radioactive counts were obtained using an Atom Lab 100 dose calibrator (Biodex). Mice were retroorbitally bled using 10 µl capillary tubes (Drummond) and radioactive counts (c.p.m.) obtained by gamma counting (Perkin Elmer). To determine total serum IgG levels, mice were retroorbitally bled with 44.7 µl heparinized capillary tubes (VWR International) and IgG concentrations in 1:25,000 dilutions of serum in PBS quantified by enzyme-linked immunosorbent assay[1]. All radioactive counts were expressed as the percentage of the levels obtained from serum samples and whole-body counting immediately before Seldeg/control delivery.

To determine the β-phase half-lives of MOG-WT or -Seldeg, HER2-WT or -Seldeg, and Abdeg (MST-HN), radiolabelled WT/Seldeg or Abdeg were injected ($n = 5$ mice per group) at a dose per mouse of 20 µg radiolabelled protein and 10, 31 and 40 µg of unlabelled proteins for MOG-WT or -Seldeg, HER2-WT or -Seldeg, and Abdeg, respectively. The β-phase half-lives for the proteins were determined by fitting the data to a biexponential model[25].

**Statistical analyses.** Statistical analyses was carried in GraphPad Prism (GraphPad Software, Inc.) by two-way analysis of variance with Tukey's multiple comparison test.

**Data availability.** All relevant data and sequences of constructs used in this study are available on request.

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

## Acknowledgements

We are indebted to Drs Jeffrey Ravetch and Patrick Smith for generously providing us with mice transgenic for human FcγRs. The HMEC-1 cell line was a generous gift from Francisco Candal (CDC). We thank Joseph J. Heimann and Keerthivasan Ambigapathy for help with plasmid generation and transfections and Kyle Current for maintaining the mouse colony. This work was supported in part by the Cancer Prevention and Research Institute of Texas (RP160051, awarded to E.S. Ward).

## Author contributions

S.C.D., R.J.O. and E.S.W. conceived this study and designed the experiments. S.C.D., P.K. and D.K.C. performed the experiments. S.C.D., R.J.O. and E.S.W. wrote the manuscript.

## Additional information

**Competing interests:** The authors declare no competing financial interests.

