## [Peer Review File · Nature Communications]

Reviewers' comments:

Reviewer #1 (expert in FnRn) (Remarks to the Author):

Comments to the authors:

The manuscript entitled, "Engineered clearing agents for the selective depletion of antigen-specific antibodies" by Devanaboyina, et al., describes the development and preliminary efficacy studies of a highly innovative new class of biologic molecules, named "seldegs" with potential application as human therapeutics in numerous indications. The authors hypothesize that circulating pathogenic antibodies that promote pathology can be selectively degraded by administering a fusion protein consisting of the target epitope of the pathogenic antibody, fused to an engineered IgG Fc fragment with enhanced binding to the neonatal Fc receptor (FcRn). The manuscript describes experiments employing a previously described Fc variant (called "Abdeg") that binds FcRn at neutral and acidic pH, whereas FcRn normally binds only at pH <6, which they denote with the term "seldeg". The manuscript describes production of two such seldegs, one called "MOG-Seldeg" targeting anti-MOG antibodies involved in the pathogenesis of multiple sclerosis, and another called "HER2-Seldeg" targeting anti-HER2 antibodies important in some types of cancer as a means of detoxification. The authors show in mouse models a significantly increased serum and whole body elimination of either anti-MOG or -HER2 radiolabeled antibodies when the respective Seldegs are administered, compared to an antigen-conjugated wild type Fc fragment, or administration of the Abdeg alone of the other Seldeg, demonstrating specificity in pathogenic antibody clearance. They next delineate the mechanism of increased clearance by showing increased uptake of target antibodies into human endothelial cells, and specifically into the endothelial cell lysosomes, a major site of IgG degradation and clearance. This is a landmark study and of great importance.

Our comments are mostly minor, and are as follows to enhance the publication.

Major comments:

1. Figure 1 shows a rapid initial drop in serum and whole body levels of radiation, as hypothesized. Subsequent experiments demonstrate trafficking of target antibodies to lysosomes. However, the change in radiation over time data also suggest an extremely prolonged terminal elimination half-life of the Seldeg-associated radioactivity, yet the authors did not address any potentially significant biologic consequences of this effect of the engineered FcRn affinity of the Seldegs. Can this be commented upon?

Minor comments:

1. Figure 1

a. Concentration-time curves (or surrogates like radioactivity remaining, as in this case) are more traditionally and intuitively displayed on a semilog format, with the concentration data plotted on a log₁₀ scale.

2. Supplemental Table 1.

a. Equilibrium K_d are provided, which, according to the methods section, were obtained by modeling surface plasmon resonance (SPR) binding curves. These data would be more completely represented if the SPR binding curves were shown.

3. Supplemental Figure 1.

a. Localization of pathogenic antibodies in lysosomes is shown. The methods section describes enzyme-linked immunosorbent assay (ELISA) determination of total IgG levels in Seldeg-treated mice. However, these data are an important control, and while control data from Fig. 1 imply the result, it would be appropriate to show that total IgG levels are unchanged by Seldeg administration, since the main premise of the paper is "selective degradation" of antigen-specific antibodies, which implies preservation of other IgG not specific for the antigen of interest.

4. The authors should cite the first paper describing monomeric Fc-fusion proteins as a prototype for the molecules described (Bitonti A, PNAS 2004).

Reviewer #2 (expert in antibody engineering) (Remarks to the Author):

Authors present a novel class of antibody-based reagents, "Seldegs", which are able to induce depletion of antigen-specific antibodies via selective degradation. They are composed of a monomeric antigen fused with a human IgG1 Fc that is modified for the higher affinity and reduced pH-dependent interaction with human FcRn using MST-HN mutations and its heterodimerization is achieved via TF/HA mutations.

The presented molecules are able to induce potent and rapid selective clearance of the antigen-specific antibody, which could be of use in treatment of antibody-mediated autoimmunity and in diagnostic imaging, albeit several antibody- and alternative recognition scaffold based reagents with a shorter than a full length IgG or tailored half-life are under development. The modifications in the designed Fc are very important as it is shown that the antigen fusions with a wild-type Fc do not elicit the same biological response. Cellular mechanism underlying the clearance is elucidated. The manuscript is clear and concise, proper controls are included. Relevant literature is cited. The Fc used for the fusion protein is substantially modified incorporating mutations which could impair its stability. I would recommend including an item of information describing biophysical characterization of the novel molecules such as the production yields, purity, stability and their monomeric character (if possible after a storage period), even if it is included in the Supplementary Material and even if these properties will strongly depend on the nature of antigen moiety fused to the Fc. This would be an important contribution to the estimation of the future manufacturability of such agents, especially as the authors propose their use in medical applications. As the authors mention themselves (line 44-50), the maintenance of monomeric status is important to prevent crosslinking, which would also cause endosomal internalization in the case of TzB.

In the abstract the term "a novel class of antibody-based reagents" instead of "antibodies" would be more concise as the described format strongly deviates from a classical antibody.

We thank the reviewers for their careful review and positive comments (e.g. ‘This is a landmark study and of great importance’). We have addressed their concerns in a point-by-point response below:

Reviewer #1:

1. *‘Figure 1 shows a rapid initial drop in serum and whole body levels of radiation, as hypothesized. Subsequent experiments demonstrate trafficking of target antibodies to lysosomes. However, the change in radiation over time data also suggest an extremely prolonged terminal elimination half-life of the Seldeg-associated radioactivity, yet the authors did not address any potentially significant biologic consequences of this effect of the engineered FcRn affinity of the Seldegs. Can this be commented upon?’*

We thank the reviewer for raising this point. In the experiments shown in Figure 1, greater than 80% of the targeted radiolabeled antibodies is cleared from the blood (and 70-75% from the whole body) within 50 hours of Seldeg delivery, and the remaining antibodies persist with half-lives that are similar to those in control (e.g. PBS-treated) mice. To investigate possible reasons for this behavior, we have carried out pharmacokinetic analyses of the Seldegs and observe that their β -phase half-lives are 47.1 hours (MOG-Seldeg) or 38.3 hours (HER2-Seldeg). These relatively short half-lives are consistent with the observations made by us and others for engineered antibodies that have increased affinity for FcRn binding at near neutral pH. The most likely explanation for the behavior of the targeted antibodies is therefore that the rapid clearance of Seldegs, combined with incomplete ‘capture’ of the pool of targeted antibodies following a single bolus injection of Seldegs, results in incomplete depletion. We have inserted this additional pharmacokinetic data as Supplementary Figure 4 and included text in the Results section to explain these points.

Minor comments:

1. *‘Figure 1. Concentration-time curves (or surrogates like radioactivity remaining, as in this case) are more traditionally and intuitively displayed on a semilog format, with the concentration data plotted on a log₁₀ scale.’*

We have redrawn the figures using a semilog format.

2. *‘Supplemental Table 1. Equilibrium K_d are provided, which, according to the methods section, were obtained by modeling surface plasmon resonance (SPR) binding curves. These data would be more completely represented if the SPR binding curves were shown.’*

We have included the SPR sensorgrams as Supplementary Figure 1 to support the affinity measurements.

3. *‘Supplemental Figure 1. Localization of pathogenic antibodies in lysosomes is shown. The methods section describes enzyme-linked immunosorbent assay (ELISA) determination of total IgG levels in Seldeg-treated mice. However, these data are an important control, and while control data from Fig. 1 imply the result, it would be appropriate to show that total IgG levels*

are unchanged by Seldeg administration, since the main premise of the paper is “selective degradation” of antigen-specific antibodies, which implies preservation of other IgG not specific for the antigen of interest.’

We have presented the data (Supplementary Figure 3) to demonstrate that Seldeg delivery does not result in significant changes in total IgG levels in the sera of mice.

4. *‘The authors should cite the first paper describing monomeric Fc-fusion proteins as a prototype for the molecules described (Bitonti A, PNAS 2004).’*

We thank the reviewer for this comment and have inserted the reference to Bitonti and colleagues.

Reviewer #2:

1. *‘I would recommend including an item of information describing biophysical characterization of the novel molecules such as the production yields, purity, stability and their monomeric character (if possible after a storage period), even if it is included in the Supplementary Material and even if these properties will strongly depend on the nature of antigen moiety fused to the Fc. This would be an important contribution to the estimation of the future manufacturability of such agents, especially as the authors propose their use in medical applications. As the authors mention themselves (line 44-50), the maintenance of monomeric status is important to prevent crosslinking, which would also cause endosomal internalization in the case of TZB.’*

We thank the reviewer for raising the important point concerning the manufacturability of the Seldegs used in the current study. The yields of the proteins are ~50 mg/L (MOG-Seldeg) and ~15 mg/L (HER2-Seldeg). In addition, both SDS-PAGE and HPLC analyses indicate that they are > 95% pure and do not aggregate following purification. The SDS-PAGE and HPLC data have been inserted as Supplementary Figure 2.

We have also assessed the stability of the Seldegs following storage at 4°C for 30 days or incubation at 37°C in serum or buffer (phosphate buffered saline) for 3-5 days. Under these conditions, MOG-Seldeg shows no signs of aggregation or degradation (SDS-PAGE, immunoblotting and HPLC analyses). Following incubation at 37°C for 5 days, there is also no indication of degradation of HER2-Seldeg (SDS-PAGE and immunoblotting) and around 2% is aggregated (HPLC). For the HER2-Seldeg there is a minor amount of degradation following storage for 30 days at 4°C (SDS-PAGE). Both Seldegs retain their binding affinities for HER2- or MOG-specific antibodies (by surface plasmon resonance using BIAcore) following incubation for five days at 37°C. These data indicate that the Seldegs have favorable biophysical properties for manufacture, and have been presented as Supplementary Figure 2 with appropriate text in the Results section.

2. *‘In the abstract the term “a novel class of antibody-based reagents” instead of “antibodies” would be more concise as the described format strongly deviates from a classical antibody.’*

We have changed the text as requested.

REVIEWERS' COMMENTS:

Reviewer #2 (Remarks to the Author):

The authors have addressed open questions by including novel experimental evidence into the manuscript and have taken into account minor changes proposed, so I can recommend the manuscript for publication.